# Sedentary Behavior Impacts on the Epigenome and Transcriptome: Lessons from Muscle Inactivation in *Drosophila* Larvae

**DOI:** 10.3390/cells12192333

**Published:** 2023-09-22

**Authors:** Avivit Brener, Dana Lorber, Adriana Reuveny, Hila Toledano, Lilach Porat-Kuperstein, Yael Lebenthal, Eviatar Weizman, Tsviya Olender, Talila Volk

**Affiliations:** 1Pediatric Endocrinology and Diabetes Institute, Dana-Dwek Children’s Hospital, Tel Aviv Sourasky Medical Center, Affiliated with the Faculty of Medicine, Tel Aviv University, Tel Aviv 6997801, Israel; avivitb@tlvmc.gov.il (A.B.); yaelleb@tlvmc.gov.il (Y.L.); 2Department of Molecular Genetics, Weizmann Institute of Science, Rehovot 7610001, Israel; dana.lorber@weizmann.ac.il (D.L.); adriana.reuveny@weizmann.ac.il (A.R.); tsviya.olender@weizmann.ac.il (T.O.); 3Department of Human Biology, Faculty of Natural Sciences, University of Haifa, Haifa 3498838, Israel; hila@univ.haifa.ac.il (H.T.); lporat@univ.haifa.ac.il (L.P.-K.); 4G-INCPM, Weizmann Institute of Science, Rehovot 7610001, Israel; eviatar.weizman@weizmann.ac.il

**Keywords:** aging, epigenetics, lncRNA, proteostasis, sarcopenia, ubiquitination, *Drosophila* larvae

## Abstract

The biological mechanisms linking sedentary lifestyles and metabolic derangements are incompletely understood. In this study, temporal muscle inactivation in *Drosophila* larvae carrying a temperature-sensitive mutation in the *shibire* (*shi^1^*) gene was induced to mimic sedentary behavior during early life and study its transcriptional outcome. Our findings indicated a significant change in the epigenetic profile, as well as the genomic profile, of RNA Pol II binding in the inactive muscles relative to control, within a relatively short time period. Whole-genome analysis of RNA-Pol II binding to DNA by muscle-specific targeted DamID (TaDa) protocol revealed that muscle inactivity altered Pol II binding in 121 out of 2010 genes (6%), with a three-fold enrichment of genes coding for lncRNAs. The suppressed protein-coding genes included genes associated with longevity, DNA repair, muscle function, and ubiquitin-dependent proteostasis. Moreover, inducing muscle inactivation exerted a multi-level impact upon chromatin modifications, triggering an altered epigenetic balance of active versus inactive marks. The downregulated genes in the inactive muscles included genes essential for muscle structure and function, carbohydrate metabolism, longevity, and others. Given the multiple analogous genes in *Drosophila* for many human genes, extrapolating our findings to humans may hold promise for establishing a molecular link between sedentary behavior and metabolic diseases.

## 1. Introduction

Sedentary behavior, characterized by low energy expenditure while sitting, reclining, or lying down [1], has been increasingly recognized as having adverse effects upon health outcomes, which may not be fully mitigated by exercise alone [2,3]. The detrimental metabolic effects of a sedentary lifestyle extend beyond the energy imbalance leading to weight gain and obesity, with muscle inactivity playing a significant role in their development [4]. Even in the absence of obesity, young patients with sarcopenia due to muscle atrophy demonstrate early-onset insulin resistance due to muscular atrophy [5]. Furthermore, physical inactivity can rapidly induce insulin resistance and microvascular dysfunction in otherwise healthy individuals, preceding the development of obesity [6]. However, the precise pathways linking sedentary behavior to insulin resistance are not fully understood and cannot be attributed solely to a decrease in muscle mass. While extensive research has focused upon identifying the multiple biological pathways activated by exercise [7,8,9,10], there are limited data on the consequences of a sedentary lifestyle on those pathways in otherwise healthy individuals.

Dynamics in epigenetics, the modifications of chromatin, which affect gene transcription without altering the genetic code, could provide mechanistic explanations for the earlier onset of diseases induced by a changing environment and behavior [11]. For example, methylation dynamics were found to evolve throughout a lifespan, with a decrease in global DNA methylation occurring with aging [12], as well as in age-related metabolic disease [13]. Various histone methylation patterns have also been described in association with aging [14]. In humans, exercise was reported as a trigger for epigenetic modifications of the DNA in muscle cell nuclei [15,16,17]. However, investigating the influence of sedentary behavior on epigenetics and gene transcription, particularly during the early stages of life, necessitates a more sophisticated approach that aims to minimize factors such as inflammatory responses and malnutrition, which can confound the interpretation of results, especially in the context of debilitating medical conditions. The present body of literature underscores the necessity for research aimed at addressing these gaps [18].

In this study, we utilized *Drosophila* larvae as an animal model for investigating the outcome of temporal muscle inactivation as a paradigm of sedentary behavior in an otherwise healthy young organism. *Drosophila* larvae were chosen based upon their advantageous genetic traits and their well-established record in muscle research [19,20]. Both the epigenetic and transcriptomic landscapes were analyzed in the skeletal muscles of inactive and active crawling larvae. Our results indicated significant differences between the experimental and the control groups, and suggested motility-dependent transcription of genes associated with muscle function, carbohydrate metabolism, longevity, downstream elements in insulin signaling, and others.

## 2. Methods

### 2.1. Fly Stocks and Husbandry

The following stocks were used: *shi^1^*; yellow, white (*yw*); *tubP-GAL80^ts^/TM2* (FBst0007017); *GAL4-Mef2.R* (FBst0027390) (obtained from Bloomington stock center, Bloomington, IA, USA); *Tub-Gal80^ts^*; *Mef2-Gal4* line (obtained from F. Schnorrer IBDM, Marseille, France). Fly lines for targeted DamID (obtained from A. Brand, Gurdon Institute, University of Cambridge, Cambridge, UK): UAS-LT3-Dm (FBtp0095492) and UAS-LT3-Dm-RpII215 (FBtp0095495) [21]. Flies carrying the *shibire^1^* allele were combined with either UAS-Dam or UAS-Dam-Pol II using standard double balancers for the X and III chromosome and for the cross with *Tub-Gal80^ts^* and *Mef2-Gal4* flies. Hemizygous *shi* males were used for all the experiments. All crosses were carried and maintained at either 25 °C, 18 °C, or 30 °C and raised on cornmeal agar.

### 2.2. Induction of Larvae Immobilization

Larvae immobilization was induced in *shi^1^* flies containing a homozygote mutation in the *shibire* gene (analogue to the human gene dynamin). The exposure of these larvae to a temperature above a known threshold (~29 °C in this line) results in a profound decrease in muscle contraction due to a robust reduction in neuromuscular junction vesicular recycling without affecting other systems, especially when the temperature shift takes place at the third instar larval stage, where normal embryogenesis and larval development are similar to the control [22].

Control (*yw*) and *shi* flies were raised on cornmeal yeast agar at 18 °C. Third instar larvae were divided into two environmental temperature groups (18 °C and 30 °C) for antibody staining and for movement analysis on the day of the experiment, and were maintained on moist Sylgard cast according to the manufacturer’s instructions (Dow Corning) without food for six hours. Fly bottles of each of the four groups (see below) were raised at 18 °C and then transferred to 30 °C for an additional 6 h for the DamID analysis.

### 2.3. Larval Locomotion Assay

Third instar larvae from all experimental groups were allowed to crawl on a flat plastic surface. The motion of each larva was recorded once for 10 s by a Samsung Galaxy S21 smartphone video camera placed on a tripod at a height of 9.5 cm parallel to the surface. The MP4 files were downloaded and read with the OpenCV (Home—OpenCV. https://opencv.org) Python package into an array. Each frame was converted to a TIFF file for motion analysis by Arivis Vision4D 4.0 software. Only frame sequences in which the larvae moved along a straight line without bending or turning were considered. Larval images were segmented by a simple intensity threshold, and larval length in each image was set as the length of the long axis of a two-dimensional-oriented bounding box. The length-adjusted motion range of each larva was calculated as the difference between its maximal and minimal length divided by the maximal length obtained during imaging.

### 2.4. Antibody Labeling of Myonuclear Epigenetic Marks in Larvae

The control and *shi* larvae were pinned, dissected, and fixed after six hours at 18 °C/30 °C, as previously described. Quantitative immunofluorescence of antibody staining against epigenetic marks was performed as previously described [23]. Briefly, dissected larva body walls were fixed in paraformaldehyde (4% from 16% stock of electron microscopy grade; Electron Microscopy Sciences, 15,710) for 20 min, washed several times in PBS with 0.1% TritonX-100 (PBST), and mounted in Shandon Immu-Mount (Thermo Fisher Scientific, Waltham, MA, USA). Fixation and antibody staining of YW and *shi* larvae were carried out in the same tube, marking one group by head excision. The following primary antibodies were used: rabbit anti-H3K9ac (Abcam, AB4441), mouse anti-H3K27me3 (Abcam 6002), and mouse anti-lamin C (LC.28.26, obtained from the Developmental Studies Hybridoma Bank, created by the NICHD of the NIH and maintained at the University of Iowa, Department of Biology, Iowa City, IA, USA). Sequential labeling of the two mouse antibodies (anti-lamin C and anti-H3K27me3) was performed as follows: dissected larvae were fixed, washed, and blocked (as above); mouse anti-H3K27me3 was added to the blocking solution for two hours, then washed, incubated with the secondary antibody, and washed again. In parallel, mouse anti-lamin C antibody (1 µg) was incubated with 5 µL of Zenon component A solution for five minutes and then with 5 µL of Zenon component B solution (Zenon^TM^ Mouse IgG_1_ Labeling Kit Z-25006, Alexa Fluor 568, Invitrogen). PBS with 0.1% triton was added to the Zenon-labeled antibody mix to a final 1:75 dilution, and then added to the dissected larvae, incubated overnight, washed, fixed as above, washed, and mounted. The following conjugated secondary antibodies were used: Alexa Fluor 555 goat anti-rabbit (Renium, #A27039) and Alexa Fluor 647 goat anti-mouse (Renium, #A21235). Hoechst 33342 (1 µg/mL; Sigma-Aldrich, St. Louis, MO, USA) was used for labeling DNA.

### 2.5. Microscopy, Image Acquisition, and Analysis

Immunofluorescence images of epigenetic marks were acquired at 23 °C on a Zeiss LSM 800 confocal microscope with a Zeiss C-Apochromat 40×/1.20 W Korr M27 lens and an Immersol W 2010 immersion medium. The samples were embedded with a high-precision coverslip of 1.5 H ± 5 µm (Marienfeld-Superior, Lauda-Königshofen, Germany) and acquired with Zen 2.3 software (blue edition). All images were taken at similar laser intensities on the same day, and the samples were incubated together in the same tube and exposed to identical antibody solutions.

Arivis Vision4D 3.1.2-3.4 was used for image visualization and analysis. Quantitative immunofluorescence analysis was performed with a dedicated pipeline that automatically segmented multiple nuclei per stack in three dimensions, and by using the denoising and Otsu threshold operation on the lamin channel. Nuclear volumes and total fluorescent intensities of the epigenetic marks and DNA were calculated by means of Arivis and exported for further analysis.

### 2.6. Targeted DamID

DNA adenine methyltransferase identification (DamID) enables the investigation of genome-wide protein–DNA interactions, reflecting transcriptional activity. We drove the expression of Dam-Pol II and Dam alone in control muscles and in *shi* temperature-sensitive mutant muscles by using a muscle-specific *mef2-gal4* driver combined with *tub-gal80^ts^*. A total of 4 genotypes were analyzed for DamID binding to DNA: (1) control larvae expressing Dam alone, (2) *shi* mutant expressing Dam alone, (3) control larvae expressing Pol II-Dam, and (4) *shi* mutant expressing Pol II-Dam. For the Dam-binding analysis, the genes identified in (3) were subtracted from the genes identified in (1) to eliminate non-specific Dam binding in the control muscles, and the genes identified in (4) were subtracted from the genes identified in (2) to eliminate non-specific Dam binding in the *shi* mutants. Finally, the resulting genes in the control muscles and in the *shi* mutants were compared between the four groups. Each group contained three independent replicates, with 25 larvae per group. For the temporal expression of the Dam constructs and for the induction of muscle inactivity, eggs were collected over 24 h at 18 °C, and then allowed to further develop at 18 °C until the early third instar larval stage. The larvae were then kept at 30 °C for six hours to allow for both muscle-specific Dam [24] expression and for inactivating larval movement. All four groups were analyzed in parallel. *Shi* hemizygous (male) mutant or control male larvae were then dissected and stored at −80 °C until all triplicates were further processed together, following a previously described protocol [25]. Briefly, genomic DNA was extracted and digested with methylation-specific DpnI enzymes that cleave at GATC sites. A double-stranded oligonucleotide adaptor was used to ensure directional ligation. The ligation was followed by digestion with DpnII that cuts only unmethylated GATCs. Finally, a PCR primer was used to amplify adaptor-ligated sequences. These amplified sequences were deep sequenced, the results were further analyzed by a bioinformatic pipeline [24], and reads were normalized to filter out non-specific Dam binding.

Each sample was processed separately by the DamID pipeline to generate normalized ratios for each pair of samples, as indicated above (https://github.com/owenjm/damidseq_pipeline). The mean occupancy per gene was then calculated by polii script (https://github.com/owenjm/polii.gene.call). The reference genome was taken from flybase (ftp://ftp.flybase.net/releases/FB2016_02/dmel_r6.10/) and GATC sites from the DamID pipeline website. After filtering genes with a DamID false discovery rate (FDR) < 0.05 (per sample), the three replicates from each group were directed to a clustering algorithm based on pairwise correlation.

A regression on principal component analysis was performed with a z-score for each of the points around the regression line in order to determine the genes with significant occupancy in *shi* relative to the control. The z-scores represent the significance of differences in RNA Pol II binding between inactive and active muscles: positive z-scores indicate increased RNA Pol II binding, while negative z-scores indicate decreased RNA Pol II binding at specific genomic loci. The cutoff criteria for significantly altered binding to a gene were set to FDR < 0.01, z-score > 1.96 (two-tailed), and GATC sites > 1, corresponding to an approximately 95% confidence interval. Independently, binding profiles (log2-fold change normalized to Dam only) of specific genes were further visualized for control and *shi* by means of the IGV browser [26]. Gene Ontology Enrichment analysis was performed with Metascape (http://metascape.org [9]) using gene lists from the three clusters identified by K-mean clustering, with a user-defined background composed of all genes that were identified as being occupied by DamID (a total of 7233 genes).

### 2.7. Statistical Analysis

The length-adjusted motion range of larvae (maximum length–minimum length/maximum length) was compared between genotypes and temperature groups by two-way ANOVA, followed by Tukey’s post hoc test. Quantitative immunofluorescence parameters (acetylation, methylation, and acetylation/methylation ratio) were compared between genotypes and temperature groups by means of a linear mixed-effects model. Genotype and temperature and their interaction were treated as fixed factors, while larvae and segments were treated as random factors. Mean and standard deviations of acetylation and methylation of groups were represented by error bars. The analyses were completed in R (v. 4.3.0) using “lmerTest” (v. 3.1). Post hoc contrasts for the mixed models were performed using “emmeans” (v. 1.8.5).

## 3. Results

### 3.1. Inhibition of Larval Muscle Contractions in Shibire Mutants

*Drosophila* larval crawling action depends upon reiterated somatic musculature contractions. We used temporally paralyzed *Drosophila* larvae as an experimental animal model for human sedentary behavior. Arrest of larval muscle contraction was obtained with homozygous *shibire*^1^ (*shi*) temperature-sensitive mutant larvae that developed normally at a permissive temperature and were then shifted to a restrictive temperature, thereby inducing temporary and reversible paralysis (Appendix A). *Shi* (dynamin-like GTPase) function at this stage (late third instar larva) has been mainly implicated in contributing to neuromuscular junction vesicular recycling, without affecting axonal conduction and muscle membrane excitability [27,28,29].

To quantify the outcome of temporal blocking of *shi* function in the restrictive temperature, we analyzed larval motility after shifting their incubation temperature from 18 °C to 30 °C and measuring their motility while crawling along a straight line 6 h later. Figure 1A,B present the change in larval length (in millimeters), reflecting larval longitudinal somatic musculature contractions over time for each larva in four study groups: (1) control wild-type larvae at 18 °C (*n* = 7), (2) control wild-type larvae at 30 °C (*n* = 8), (3) *shi* homozygous mutant larvae at the permissive temperature of 18 °C (*n* = 7), and (4) *shi* homozygous mutant larvae at the restrictive temperature of 30 °C (*n* = 6).

Taken together, these measurements indicated that the extent of somatic muscle contraction in *shi* mutant larvae is similar to that of wild-type larvae when incubated at the permissive temperature, whereas the *shi* larvae are essentially immobile at the restrictive temperature. Notably, this phenotype was completely reversible: once the *shi* mutant larvae were shifted back from 30 °C to 18 °C, they exhibited normal crawling behavior within 1–2 min. 

### 3.2. Epigenetic Landscape Analysis in Myonuclei

To assess changes in the epigenetic landscape of the inactive larvae, we analyzed larvae from the four study groups following their dissection and labeling with antibodies specific for the chromatin epigenetic repression mark H3K27me3 and the epigenetic activation mark H3K9ac.

Figure 2A shows representative images of the labeled muscle nuclei. H3K9 acetylation in the inactive *shi* larvae (incubated at 30 °C for 6 h) demonstrated an enhanced punctate pattern that did not appear in either the active *shi* at 18 °C or in the control groups. Quantification of the fluorescent signals of either active or repressive epigenetic marks was performed by measuring the entire fluorescence signal of each of the epigenetic marks in the same nuclei throughout their entire volume. We analyzed 5–6 distinct larvae and all nuclei from 1–5 distinct muscle fibers per larvae for each of the four study groups (Figure 2B). The results indicated that the ratio of acetylation/methylation signals per individual muscle nucleus was significantly higher in the inactive muscles (*p* = 0.025), whereas this ratio in all control groups was essentially similar (*p* = 0.995) (Figure 2B). According to the linear mixed model, there was no significant difference in the extent of total H3K27me3 or H3K9ac fluorescent signal in each of the groups, possibly due to increased variability between larvae (Appendix A). Specifically, inactive *shi* larvae exhibited a significantly higher variability in the methylation signal, as evidenced by an increased standard deviation (*p* < 0.001) (Appendix A). Interestingly, whereas the H3K9ac-positive puncta appeared significantly more intense in the inactive muscle nuclei, indicating altered chromatin acetylation distribution (Figure 2A), the average degree of fluorescence per nuclear volume was not significantly higher. Taken together, the results indicate a significant correlation between muscle inactivation and a higher acetylation/methylation ratio as well as increased methylation variability, implying a tendency towards gene transcription stimulation.

### 3.3. Altered Genome Pol II Binding Profile to DNA in the Inactive Muscles

We further addressed potential transcription changes in the inactive muscles by investigating the binding of RNA Pol II along the genome in two experimental groups, namely *shi* or wild-type larvae, both raised at 18 °C and transferred to the restrictive temperature of 30 °C for 6 h at the third instar developmental stage. We chose the “targeted DamID” (Tada) experimental approach for the genome-wide analysis of Pol II binding, where muscle-specific inducible Dam-Pol II fusion protein binding to DNA is analyzed to bypass the need for tissue isolation of the larval–somatic musculature, and because this approach detects the initiation of transcription, namely the binding of RNA Pol II. A fly line combining a temperature-sensitive Gal 80^ts^ construct with a muscle-specific *Mef-Gal4* driver was genetically combined with the *shi* mutant allele and crossed with a fly line carrying the Dam-Pol II (or control Dam) construct. This was carried out so that both the induction of muscle inactivation in hemizygous *shi* and the induction of Dam-Pol II expression were simultaneously promoted upon transfer of their progeny third instar larvae to 30 °C.

Overall, the Dam-Pol II analysis identified 2010 genes to which Pol II was bound in muscle nuclei (Figure 3, Table 1, and Appendix A). Of these, 1734 (86.3%) were coding genes and 201 (10%) were long non-coding RNAs (lncRNAs). Alterations in Pol II binding in the inactive muscles were observed in 121 genes (Appendix A), representing 6% of all genes analyzed. Table 1 presents categories of genes that were analyzed and, among them, those which were altered in the inactive muscle. Interestingly, 36 (29.8%) out of the 121 genes that underwent alterations were lncRNAs and 59.5% were coding genes, implying a higher proportion of lncRNAs in the group of changed genes (see also Figure 3).

Figure 4A presents the difference in Pol II binding to protein-coding genes between inactive larvae (*shi*) and controls, with the genes divided into functional groups. Notably, two genes associated with lifespan, *Thor* (4E-BP) and *mir-282,* and one gene associated with DNA repair, *Pdrg1*, exhibited reduced binding to Pol II in the inactive muscles. Moreover, a group of five genes associated with protein ubiquitination (*rngo, Rpn9, Rpn13, Rpt3,* and *Ubi-p5E*) was functionally linked to proteostasis (Figure 4B), and exhibited reduced binding to Pol II in the inactive muscles as well (see Discussion). Functional grouping of the genes that were affected in the inactive muscles revealed a group of 10 genes associated with muscle function (*zasp66, l(2)elf, CG5177, TpnC73F, Tm2, wupAAld, CG45076, sqh, Act87E,* and *Cam*) which exhibited reduced Pol II binding, suggesting motility-dependent transcription of muscle genes (Figure 4A,B). Furthermore, a group of transcription regulators, including *Polr2A*, a subunit of the RNA Pol II transcription complex, exhibited reduced binding to Pol II, suggesting a muscle-specific decrease in transcription in the inactive muscles. In addition, suppressed Pol II binding to *pepck,* a key enzyme in gluconeogenesis, was observed in the inactive muscles. Taken together, these findings suggest that there is a downregulation of transcription in muscle-function genes as well as in other genes essential for muscle protein maintenance shortly after larval muscle inactivation.

## 4. Discussion

In this study, we employed a *Drosophila* experimental model to address the impact of early-life sedentary behavior on transcriptional regulation in muscle nuclei (myonuclei). The third instar larval stage, which typically spans 1–2 days, is marked by relatively limited larval growth compared to earlier stages. In a human context, this phase might be analogous to late adolescence, and 6 h for larvae might be equivalent to a few months in the timeline of human development. Our findings demonstrated that short-term muscle inactivity induces changes in the epigenetic fingerprint of myonuclei, as well as alterations in transcriptional activity. Specifically, muscle inactivity was associated with a decreased profile of Pol II-binding to muscle-specific genes, as well as genes involved in protein ubiquitination, DNA repair, longevity, and gluconeogenesis. The suppressed expression of these genes may contribute to muscle degeneration in inactive muscles and potentially contribute to the development of sarcopenia, a hallmark of aging in humans.

The observed Increase in the acetylation/methylation ratio in the inactive muscles suggests a reduction in transcriptional restraint, a phenomenon previously associated with aging [14]. Importantly, the shift in the acetylation/methylation ratio was not observed in the control groups, indicating that it is specifically linked to muscle inactivity rather than to temperature-associated changes. The alterations in the epigenetic landscape indicate a general trend towards a more open chromatin configuration, reminiscent of aging cells [30]. Such changes have been linked to transcriptional dysregulation [31], further emphasizing the potential impact of muscle inactivity upon gene expression.

Numerous studies have highlighted the crucial role of histone acetylation and deacetylation in bridging epigenetics, transcriptional regulation, and metabolism. However, the balance between histone acetylation and deacetylation has not shown a clear pattern in aging or metabolic disorders [32]. Similarly, the interplay between net acetylation and methylation of chromatin has not been extensively described in the context of aging. Interestingly, while the shift in the acetylation/methylation ratio may suggest an augmented binding of Pol II, the proportion of the genes exhibiting enhanced Pol II binding was smaller compared to genes with reduced Pol II binding, suggesting additional levels of transcription regulation [33]. The upregulated genes included *mir-10*, 13 lncRNAs, and the heat-shock protein *HSP67Ba*, which have reportedly been involved in stress response, possibly activated by the heat shock [34].

Importantly, a group of muscle-specific genes encompassing 10 genes coding for muscle structural and functional proteins exhibited reduced Pol II binding in the inactive larvae muscles. This decreased Pol II binding predicts a decline in the expression of these genes, potentially leading to impaired muscle performance and loss of muscle mass. Given the existence of analogous genes in *Drosophila* for many human genes, including muscle structure and function genes, cautious extrapolation may be made for the implications of temporary muscle inactivity experienced in young age. Our findings raise a concern that early onset sedentary behavior can expedite the onset of sarcopenia, a hallmark of aging [35]. 

In addition, sarcopenia is closely associated with disrupted glycemic homeostasis, and a bidirectional causal relationship exists between the two conditions [36]. Furthermore, in our study, muscle inactivity correlated with altered binding of Pol II to genes associated with the metabolism of carbohydrates. Of those, the suppressed Pol II binding to *pepck,* a key enzyme in gluconeogenesis, aligns with expectations, since reduced metabolism resulting from muscle inactivity requires less fuel [37]. An upregulation was observed in Pol II binding of the glucose transmembrane transporter (encoded by the gene *MFS3*), which facilitates insulin-promoted utilization of plasma-derived carbohydrates in *Drosophila* and humans. The net effect expected from suppressed glucose synthesis combined with augmented glucose intracellular internalization is a decrease in blood (or the equivalent fly hemolymph) glucose levels. However, since insulin sensitivity is tissue-specific [38], and given that our study specifically targeted Pol II binding in myonuclei, we were unable to assess the impact of muscle inactivity on the transcriptome of other tissues involved in glycemic regulation and to evaluate implications for organism-wide insulin sensitivity versus resistance.

*Thor*, a key gene in the determination of insulin sensitivity, was also reduced in the Dam-Pol II analysis of inactive muscles. The *Thor* gene encodes the target protein of the transcription factor FOXO and promotes pathways of intracellular protein degradation of damaged protein aggregates, thereby preventing cell degeneration [39]. Demontis et al. [40] reported that intra-muscular signaling by the *FOXO/Thor* pathway had an organism-wide effect through insulin homeostasis regulation: specifically, its activation delayed the age-related accumulation of protein aggregates in other tissues, including muscles. *FOXO/Thor* signaling was found to prevent intracellular protein buildup, at least in part by promoting the activity of the autophagy–lysosome system that degrades ubiquitinated proteins [40]. It follows, therefore, that the observed downregulation of *Thor* transcription in inactive muscles might impair its potential beneficial effect on muscle growth.

Consistent with motility-dependent maintenance of normal proteostasis, we identified five genes involved in the protein ubiquitination pathway (*rngo, Rpn9, Rpn13, Rpt3,* and *Ubi-p5E*) that were downregulated, and one gene, *Roc1a*, that was upregulated. Harmful proteins undergo ubiquitination and are immediately degraded by the proteosome pathway in healthy individuals with normal proteostasis [41]. Any disruption to this protein clearance system can result in the accumulation of detrimental protein aggregates, ultimately leading to the onset of various diseases, including neurodegenerative conditions [42]. The accumulation of damaged proteins that stems from reduced ubiquitination is associated with cellular degeneration in age-related diseases [43] and contributes to the development of metabolic disorders [44].

Finally, our analysis indicated a specific impact of muscle inactivity on lncRNAs, which were three times more prevalent among the genes affected by muscle inactivity compared to their representation in all genes. LncRNAs play an important role in chromatin remodeling and in the regulation of transcription by binding promoters or enhancers and either activating or repressing transcription [45]. Evolutionarily conserved lncRNAs have been implicated in aging and cancer [46], and quantitative trait shifts in the expression of lncRNAs have been observed in diabetes type 1 and 2, as well as in coronary artery disease [21]. Since the precise functions of the lncRNAs identified in our study remain unclear, it is impossible to draw definitive conclusions about their specific metabolic implications at this time.

We acknowledge limitations in this study that bear mention. The effects of temporary muscle inactivation might not fully replicate the habitual decline in activity in humans with sedentary lifestyles. An additional constraint stems from the considerable evolutionary gap between flies and humans, which restricts our ability to extrapolate findings to metabolic disorders in humans. Nevertheless, extensive research has established the genetic resemblance between *Drosophila* and humans [47], underscoring that our innovative findings lay the foundation for future investigations.

## 5. Conclusions

Our analyses reveal the impact of a relatively short period of muscle inactivity on chromatin regulation, encompassing shifts in epigenetic balance and diverse modifications in transcriptional machinery. Of note is the primary impact of muscle inactivation on three major pathways, namely longevity, glycemic regulation, and protein proteostasis through the ubiquitin pathway. Extrapolating these findings to humans holds the potential of establishing a molecular connection between sedentary behavior and disease, such as type 2 diabetes, as well as premature aging.

## Figures and Tables

**Figure 1 cells-12-02333-f001:**
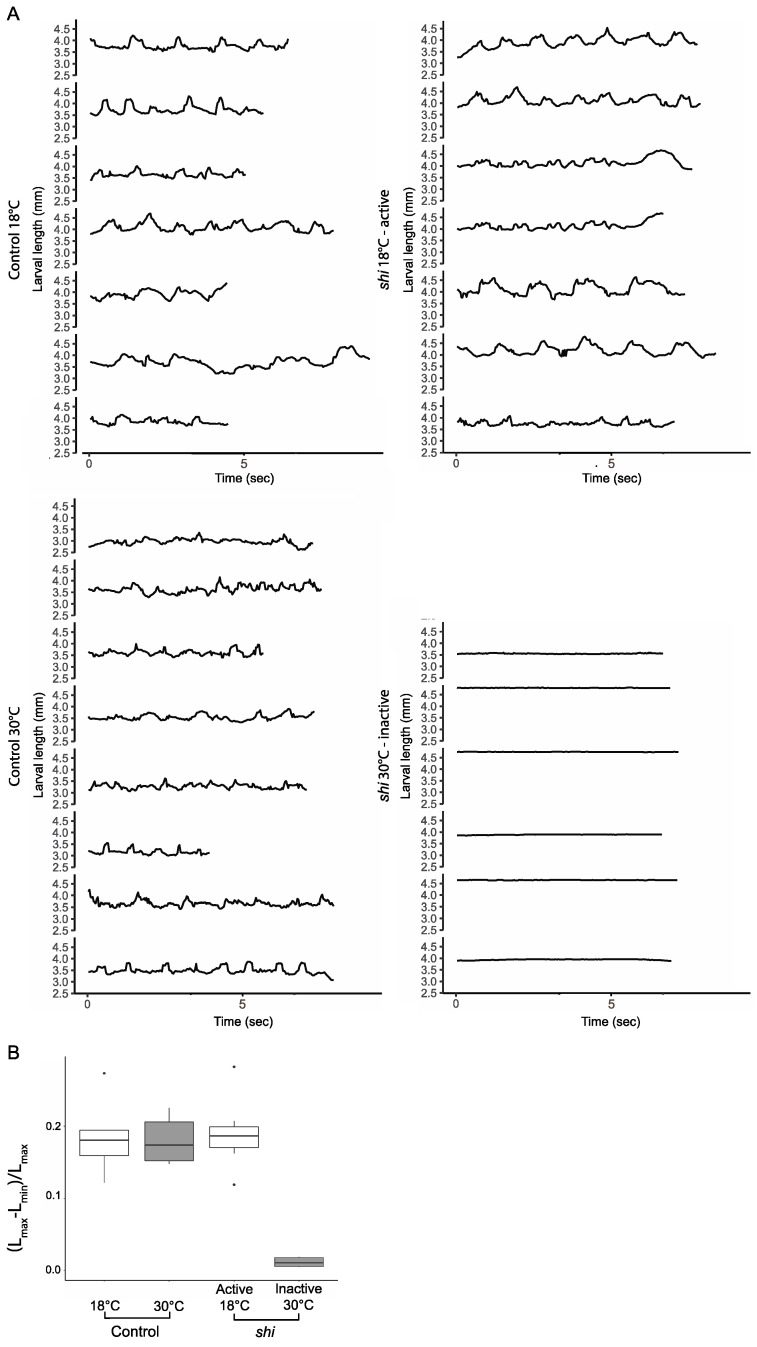
**Motion analysis of active and non-active *shi* mutant larvae.** (**A**) Larval length over time: control larvae at 18 °C (upper left) and at 30 °C (lower left), as well as *shi* at 18 °C (permissive temperature, upper right) and at 30 °C (restrictive temperature, lower left). (**B**) Quantification of the larval motion range in each group in (**A**), calculated as the maximal larval body length change during crawling normalized by its maximal length (Lmax − Lmin/Lmax). Larval body motion range was significantly lower in the *shi* larvae at 30 °C compared to the *shi* mutant larvae at 18 °C (0.011 ± 0.006 vs. 0.190 ± 0.050, *p* < 0.001; *p* < 0.001). There were no significant differences between the *shi* mutant at the permissive temperature (18 °C) and the two control groups at 18 °C and at 30 °C (0.180 ± 0.032 and 0.183 ± 0.047, *p* = 0.990 and *p* = 0.958, respectively), indicating their normal motility at both temperatures.

**Figure 2 cells-12-02333-f002:**
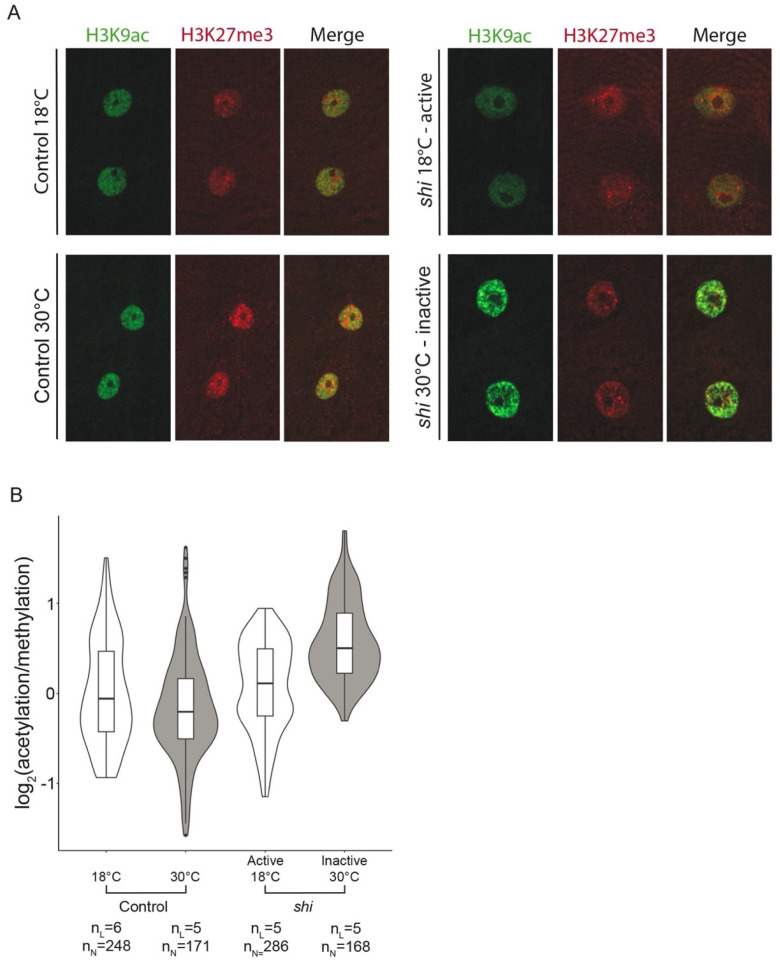
**Epigenetic landscape analysis of myonuclei in larvae with active and inactive muscles.** (**A**) Representative images of the muscle nuclei labeled with anti-H3K9ac (green) and H3K27me3 (red) of control larvae at 18 °C (upper left panel) and 30 °C (lower left panel) and *shi* larvae at 18 °C (upper right panel) and 30 °C (lower right panel). (**B**) H3K9ac/H3K27me3 ratio in the muscle nuclei of larvae in the four study groups (n_L_ = number of larvae, n_N_ = number of nuclei). A linear mixed-effects model demonstrated a significantly higher acetylation/methylation ratio in the inactive muscles (*shi* 30 °C) (*p* = 0.025), whereas this ratio in all control groups was essentially similar (*p* = 0.995).

**Figure 3 cells-12-02333-f003:**
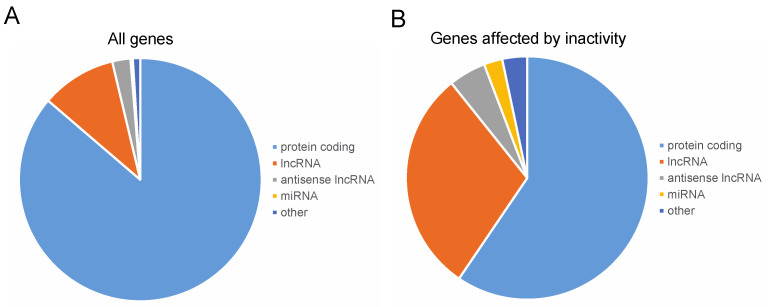
**Classification of genes showing altered Dam-Pol II binding in the inactive muscles.** (**A**) The proportion of gene classifications that underwent Pol II binding analysis. (**B**) The proportion of gene classifications that underwent alterations in Pol II binding in the inactive versus active muscles. Overall, 2010 genes were bound specifically to Pol II. Coding genes represented 86.3% of the total and long non-coding RNAs (lncRNAs) represented 10%. Pol II binding in the inactive muscles was altered in 121 genes (6%), but the proportion of genes coding for long non-coding RNAs (lncRNAs) was three-fold higher (29.8%).

**Figure 4 cells-12-02333-f004:**
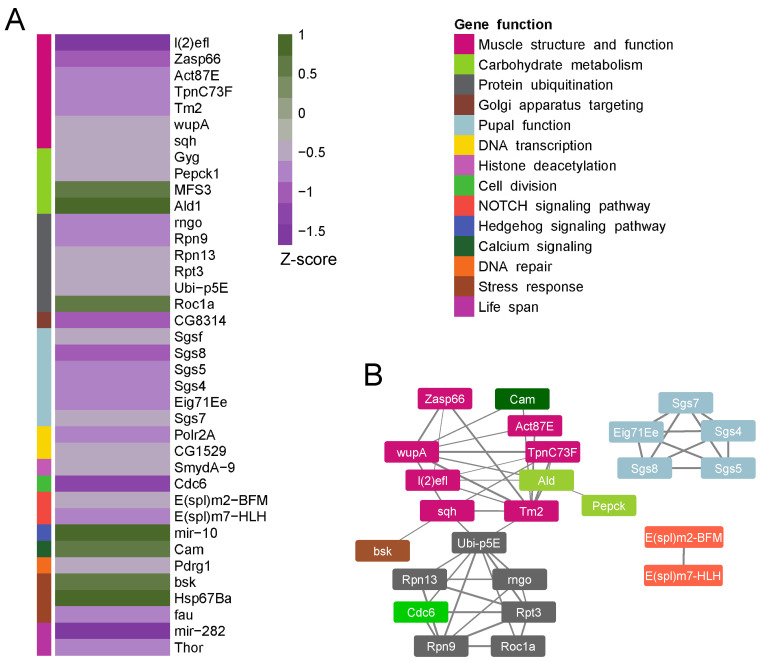
**Functional classification of the protein-coding genes altered in the inactive muscles.** (**A**) The degree of altered Pol II binding to functionally based selected genes in inactive versus active muscles, with color coding of functional groups. The z-scores represent the significance of differences in RNA Pol II binding between inactive and active muscles: positive z-scores indicate increased RNA Pol II binding, while negative z-scores indicate decreased RNA Pol II binding. (**B**) String analysis of the genes presented in (**A**) with their color-coded functional grouping.

**Table 1 cells-12-02333-t001:** Categories of genes and their proportions analyzed by DamID protocol and altered by muscle inactivity. Data are presented as number (%). LncRNA, long non-coding ribonucleic acid; snoRNA, small nucleolar RNA; miRNA, micro RNA; snRNA, small nuclear RNA; tRNA, transfer RNA.

Type of Gene	Analyzed Genes	Genes Affected by Muscle Inactivity
Total, *n*	2010	121
Protein coding	1734 (86.3)	72 (59.5)
lncRNA	201 (10)	36 (29.8)
antisense lncRNA	49 (2.4)	6 (5)
snoRNA	8 (0.4)	1 (0.8)
miRNA	6 (0.3)	3 (2.5)
snRNA	7 (0.3)	1 (0.8)
Pseudogene	3 (0.2)	1 (0.8)
tRNA	2 (0.1)	1 (0.8)

## Data Availability

The data that support the findings of this study are available from the corresponding author upon reasonable request.

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
