# Peer review of "Sedentary Behavior Impacts on the Epigenome and Transcriptome: Lessons from Muscle Inactivation in Drosophila Larvae"

_cells, 2023, doi:10.3390/cells12192333_

Round 1

Reviewer 1 Report

This work uses Drosophila larvae with a temperature-sensitive mutation in the shibire1 (shi) gene to mimic sedentary behavior during early life and investigate its impact on transcriptional outcomes. The paper is clear in its objective, which is to understand the biological mechanisms connecting sedentary lifestyles and metabolic disorders. The authors found changes in both epigenetic and genomic profiles of RNA Pol II binding in inactive muscles compared to controls. This suggests a comprehensive approach to examining the effects of muscle inactivity.

The paper presents quantitative data regarding the number of genes affected by muscle inactivation (121 out of 2010 genes), with a particular focus on genes related to long-term health, DNA repair, muscle function, and proteostasis. The work shows that muscle inactivation leads to alterations in the epigenetic balance of active and inactive marks. This is a critical finding as it may shed light on how sedentary behavior affects gene regulation. The text concludes by suggesting that the findings may have implications for understanding the molecular link between sedentary behavior and metabolic diseases in humans.

This paper can be published. There are only several small concerns:

1) It would be nice to present both Fig. 1B and Fig. 2B in a similar format for easier comparison (now we see the box-and-whisker plot in Fig. 1B and the violin plot in Fig. 2B).

 2) The second problem is conceptual. The authors said in the Abstract (and a similar sentence in the Discussion) "Extrapolating these findings to humans holds promise for establishing a molecular link between sedentary behavior and metabolic diseases." However, the evolutionary distance between the fly and the human in too large to make such extrapolation easily. I believe that a certain discussion of possibility of such a great extrapolation (with corresponding references) and some words of caution are needed here. Furthermore, it is worth emphasizing in Discussion that the inactivation was caused by the mutation (which is hard to compare with a habitual decline of activity in humans) and the inactivation was very large (17-fold).

3) It is very interesting to know how the shibire1 (shi) mutation affects muscle cell ploidy because genome accumulation can promote chromatin opening (DOI: 10.3390/ijms23179691 ). If is possible and if only it is quite easy tho you, could you please evaluate muscle cell polyploidy (just for about 100 cells per 1 Drosophila Fly using simple Hoechst 33342 staining

 )? If also would be interesting to find out whether shibire1 (shi) mutation alters protein content in muscle cells. If it is possible, please, evaluate it. If this task is too complicated, you can leave it for your next article.

4) The Supplementary Table containing the complete list of shi  regulated genes uncovers a lot of exciting information. For example, it points to the upregulation of Myc oncogene promoting chromatin opening.  Therefore, it would be good to confirm Myc overexpression by Real-time PCR, or by immunoblotting. Please do not take this recommendation as a serious concern.  You can perform this confirmation only if it is quite easy to you.   With respect for your talent, The Reviewer

English is quite good

Reviewer 2 Report

This is an interesting paper examining epigenetic and transcriptomic changes linked to sedentary behavior. The authors model this using temperature sensitive muscle inactivation in 3rd instar Drosophila larvae and have included all the appropriate controls. The methods are interesting and for the most part, well described, and the amount of data presented are well within the scope of this journal.  Statistical analyses are appropriate for the studies performed. The genetic targets and epigenetic changes highlighted parallel what one might expect from inactivity, and novel identified candidates may be useful to future studies.

I do have a few minor concerns that I would recommend prior to publication, listed below:

1) The introduction jumps around from topic to topic, beginning with T2D, then moving to sedentary behavior, then sarcopenia and insulin resistance. While relevant to the findings presented, these are not directly studied in the manuscript. When reading the discussion these connections are much more logical and clear. I would recommend re-working the introduction to create a more logical flow of information. The primary input here is muscle inactivation/inactivity modeling sedentary behavior, and the changes in gene activity that result. The importance of these findings and their implications to disease might be better discussed after reintroducing the primary aims of the study.

2) The authors introduce the study as a model for sedentary behavior, but also frequently discuss how the muscle inactivation is short-term. The 3rd instar larval stage is quite a brief developmental period, so longer term inactivation would be very difficult.  It would aid the reader, particularly non-Drosophilists, to better justify the 6 hour inactivation window. While there is not a ratiometric relationship between Drosophila development/lifespan and humans, a brief discussion of the impact of 6 hours of immobilization may help better understand it's translatability. 

3) Methods: Line 140: Fluorescence images were taken at "similar intensities". Since quantitative immunofluorescence was used, a short description of the authors methods to ensure samples were comparable seems appropriate.

line 144: three dimension should read three dimensions
